# A qualitative examination of causal factors and parent/caregiver experiences of non-fatal drowning-related hospitalisations of children aged 0–16 years

Boshra Awan[1]*, Suzanne Wicks[1], Amy E. Peden[2,3]

1 Kids Health Promotion Unit, Sydney Children's Hospitals Network, Westmead, New South Wales, Australia, 2 School of Population Health, Faculty of Medicine and Health, UNSW Sydney, Kensington, New South Wales, Australia, 3 College of Public Health, Medical and Veterinary Sciences, James Cook University, Townsville, Queensland, Australia

* boshra.awan@health.nsw.gov.au

## Abstract

Fatal and non-fatal drowning is a significant public health issue, which disproportionately impacts children and young people. In Australia, the highest fatal and non-fatal drowning rates occur in children under five years of age. To date, little qualitative research has been conducted on non-fatal drowning, with causal factor analysis generally conducted using coronial and hospital data. This study's aim was to identify causal factors in hospital treated cases of non-fatal drowning in children as qualitatively self-reported by parents and caregivers. Cases of unintentional child (0–16 years) non-fatal drowning admissions and Emergency Department presentations to three tertiary care paediatric hospitals in New South Wales, Australia were identified via International Classification of Diseases (ICD) coding. Parents and caregivers of drowning patients were invited to participate in a semi-structured interview. Data were thematically coded using an inductive approach, with a focus on causal factors and recommendations for preventive approaches. Of 169 incidents, 86 parents/caregivers were interviewed. Children hospitalised for drowning were more often male (59.3%), aged 0–4 years (79.1%) and 30.2% were from household who spoke a language other than English. Qualitative incident descriptions were coded to five themes: lapse of supervision, unintended access (commonly in home swimming pools), brief immersion (usually young children bathing), falls into water and ongoing impacts. Drowning prevention recommendations were grouped under supervision, pool barriers and maintenance, cardiopulmonary resuscitation (CPR) training and emergency response, drowning is quick and silent, and learning swimming. Parents and caregivers of young children require ongoing education regarding supervision distractions and pool barrier compliance. Additional challenges are faced by those in rental properties with pools, parents/caregivers who cannot swim, and parents/caregivers from culturally and linguistically diverse backgrounds. Affordable, accessible, and culturally appropriate swimming lessons, water safety education and CPR training should be made more available for adult caregivers, particularly in languages other than English.

**Data Availability Statement:** Underlying data used in this study cannot be uploaded due to ethical constraints surrounding the sensitive nature of the

data. Those interested in learning more about the data and potentially accessing the data, can contact the Hunter New England Human Research Ethics Committee via email (hnehrec@hnehealth.nsw.gov.au) and quote reference number 13/05/15/4.06.

**Funding:** Author AEP is funded by a National Health and Medical Research Council (NHMRC) Emerging Leadership Fellowship (Grant ID: APP2009306).

**Competing interests:** The authors have declared no competing interests exist.

## Introduction

Drowning is a significant public health issue, which disproportionately impacts children and young people under 25 years of age [1, 2]. In Australia, an average of 273 people die from unintentional drowning each year, with the highest fatal drowning rates seen in young children under five years of age [3].

In addition to fatal drowning, drowning may also be non-fatal in outcome [4], however, significantly less is known about non-fatal drowning burden. In Australia, non-fatal drowning-related hospitalisations are also highest among children under five, a ratio of 8 hospitalisations for every fatal drowning [5].

The range of health outcomes for a person who survives a drowning incident range from no long-term effects, through to severe brain and other organ damage [6]. A child sustaining neurological deficits due to drowning is likely to experience a lifetime of significant impairment, requiring sustained support from those who care for them. Even among children who showed no apparent neurological problems on discharge after a non-fatal drowning incident, 22% showed behavioural problems, poor communication, executive function and learning difficulties at some point during their five-year follow-up [7].

Studies of non-fatal drowning have almost exclusively been quantitative in focus, with little qualitative research on the topic. In addition, detailed causal analysis of fatal and non-fatal drowning incidents in Australia rely largely on coronial and hospital data [3, 8]. Coronial files, such as a police reports, may include incident information derived from parents and caregivers, though such data are available for cases of fatal drowning only. Qualitative data can provide rich detail on non-fatal drowning incidents including valuable insights from parents and caregivers to inform future child drowning prevention efforts. To address this research gap, the current study aimed to qualitatively investigate the circumstances and causal factors for unintentional non-fatal drowning incidents in children and adolescents aged 0–16 years (henceforth referred to as children) to inform preventive approaches.

## Methods

This is a retrospective cross-sectional qualitative study with the parents/caregivers of children who have presented to three tertiary paediatric hospitals in New South Wales, Australia's most populous state.

### Study setting

Participant recruitment occurred through the three paediatric hospitals in the state of New South Wales (NSW), Australia. These are The Children's Hospital at Westmead (CHW), Sydney Children's Hospital, Randwick (SCH) and John Hunter Children's Hospital, New Lambton Heights (JHCH). The Children's Hospital at Westmead is located in the heart of Western Sydney, one of the most culturally and linguistically diverse areas and with one of the largest Aboriginal Australian populations in the state of New South Wales. The Children's Hospital at Westmead is one of the leading paediatric hospitals in Australia, therefore it services children and young people aged 0–16 years old Sydney-wide, as well as regional and remote NSW. The Sydney Children's Hospital Randwick is located in the eastern suburbs of Sydney, New South Wales, Australia. It is also one of the leading paediatric hospitals in Australia, and services children and young people Sydney-wide, as well as regional and remote NSW. The hospital is located just two kilometres from the most popular beaches of Sydney including Coogee and Bondi, and therefore services majority of drowning incidents that occur at beaches.

John Hunter Children's Hospital, New Lambton Heights is located to the West of the City of Newcastle, approximately 150 km north of Sydney and just 8 km away from the closest

**Table 1. All-cause case load for hospitals included in this study in 2021.**

| Hospital | Occasions of service for non-admitted patients | Emergency Department presentations | Hospital admissions |
|---|---|---|---|
| The Children's Hospital at Westmead (CHW) | 923,741 | 59,595 | 39,288 |
| Sydney Children's Hospital, Randwick (SCH) | 346,100 | 38,191 | 17,649 |
| John Hunter Children's Hospital, New Lambton Heights (JHCH) | 993,140 | 24,499 | 11,229 |

beach. The John Hunter adults and children's hospitals are the largest outside of the Sydney metropolitan area. John Hunter Children's Hospital primarily services children and young people aged 0–16 years from the Hunter and New England regions of NSW, as well as neighbouring areas of Mid North Coast and Northern NSW. Table 1 depicts the caseloads in the 2021 calendar year for these hospitals.

## Patient identification and eligibility criteria

Children between 0–16 years old who had presented to one of the three hospitals as a consequence of a non-intentional non-fatal drowning incident were eligible for inclusion in the study. Hospital separations due to drowning were identified where the principle diagnosis was any code in the ICD-10-AM [9] Chapter XIX Injury, poisoning and certain other consequences of external causes (S00-T98) and where the first reported external cause of morbidity was Accidental Drowning and Submersion (W65-74). Children who were admitted due to incidents which, on closer examination, were incorrectly coded as an immersion/submersion incident and were not in-fact drowning-related, were excluded from the study. Children with pre-existing health conditions or a disability were also excluded from this study. The rationale behind decision was that greater attention should be paid to this sub-group and different tools used to capture insights that increase drowning risk among this unique population. It is however, acknowledged as a limitation of this research and we recommend dedicates studies with parents/caregivers of children with a disability or pre-existing medical condition in the future.

Medical records of patients aged 0–16 years who had presented (emergency department and inpatient) to the three paediatric hospitals in NSW for a non-intentional non-fatal drowning incident were collected between 1 July 2015 and 31 March 2018. Medical records were obtained through two methods: 1) Patient records data (S1 File) completed by an on-site research team member at each of the three paediatric hospitals. The on-site research team member was usually the Clinical Nurse Consultant (CNC) of the Trauma Unit or Data Manager. Completed questionnaires were forwarded on a monthly basis to the Research Officer based at CHW; or 2) Data were retrieved directly from both hospital inpatient and emergency department medical record databases for patients who presented to CHW and SCH only. Medical records of drowning incident patients presenting at JHCH were exclusively provided by the on-site research team member.

## Recruitment and informed consent

Parents/caregivers of eligible children were provided with an information sheet and a consent form when they attended hospital. For those parents who were willing to be contacted (i.e., completed the consent form in hospital), parents/caregivers were contacted via phone by the Research Officer at CHW and again invited to participate in the study. This invitation reiterated the nature of the project and the intent for findings to be used to inform drowning prevention and water safety initiatives. Research officers were empathetic and genuine in their approach, to build rapport and trust.

Consent was also verbally obtained from the parent/caregiver of the child via telephone. When the researcher made the telephone call to the parent/caregiver, the researcher asked the question "would you like to be part of this research by answering some questions?". If they agreed then the researcher commenced asking the questions. If they declined, the researcher thanked them for their time and they ended the call. The parent/caregivers' consent was recorded on the researcher's database as "yes" or "no".

If an interviewee required a language interpreter, this was organised through the Health Care Interpreter Service, NSW Health. Two interviewees required an interpreter, both in the Arabic language. In most cases, contact and consent occurred at least three months after the child had been discharged from hospital. Each family was called three times over the span of two weeks before a voicemail message was left, and a text message sent via Message Media (if a mobile number was known) briefly explaining the purpose of the study and asking them to call back if they were interested in being involved. If the parent/carer did not respond to the one voicemail and one text message, they were placed on the 'could not be contacted' list.

## Data collection

If the parent/caregiver agreed to participate, and after informed consent had been expressed verbally, the parent/caregiver were asked questions from a semi-structured in-depth interview over the phone (S2 File). If an interviewee was willing to participate but unavailable at the specific time, an alternative time was arranged. The answers to the questions were hand-written by the Research Officer who conducted the interview on a hard copy of the questionnaire and were subsequently typed up into an Excel spreadsheet, with a random 10% check between hard copy questionnaire and Excel. Due to the sensitive nature of the subject matter, phone conversations were not recorded. All data were kept confidential in locked cupboards and password-protected Excel files.

Information obtained from the interview included the participant's demographics, circumstances surrounding the drowning, location of drowning, type of supervision, administration of CPR, swimming ability of the child, details of the swimming pool (if known), including pool barriers, council approval, known faults with the pool barrier and any recommendations or comments the interviewee would give to others to prevent a similar incident from occurring. Type of supervision was defined as: Within arm's reach—Parent/caregiver was in the water with the child and within arm's reach of them; From a distance—Parent/caregiver was not in the water but was within a few metres of the child; Unsupervised and aware—Parent/caregiver was not present but was aware that the child was in or near water; Unsupervised and unaware: Parent/caregiver was not present and was unaware that the child was in or near water.

## Data analysis and interpretation

Quantitative data about the drowning patient, parent/caregivers and incident characteristics were reported using univariate statistics. The proportion of the sample who were Aboriginal and/or Torres Strait Islander and/or spoke a language other than English at home, were compared to the broader New South Wales population using Australian Bureau of Statistics Census data [10].

Thematic analysis was conducted on the questions *"Are you able to tell me what happened to (child's name)?"* and *"Do you have any advice or recommendations you would give to another parent to prevent a similar incident from occurring?"* For these two questions, two researchers separately read through the Excel files and independently thematically coded the free text responses using an inductive methodology as outlined by Braun and Clarke [11]. This five-phase methodology was undertaken as follows: Both authors separately familiarized themselves

with the data by rereading the data and noting down initial themes (Phase 1). Both authors then separately generated initial codes by systematically coding interesting features of the entire dataset (Phase 2). Both authors then came together to cross-check and confirm individual codes and thematically determine categories (Phase 3). Separately, both authors sorted codes into categories (Phase 4). Both authors then came back together to compare the contents of each category and refine any outstanding issues (Phase 5).

### Ethics

Ethics approval was provided by the Hunter New England Human Research Ethics Committee (HREC) (HNELHD) with Reference No: 13/05/15/4.06 and NSW HREC Reference No: HREC/13/HNE/173. This study was conducted in accordance with the Helsinki Declaration of 1975, as revised in 1983.

## Results

In total, 169 children aged 0–16 years were eligible for inclusion across the study period. Due to a range of reasons, as depicted in Fig 1, in total, there were 86 parents/caregivers who consented and were interviewed for this study.

### Demographics and incident information

The demographic details of the interviewee and the 86 children who had a non-fatal drowning related hospitalisation, can be found in Table 2. Of those interviewed, 86.0% were female and the mother (83.7%). Of the children who drowned, over half (59.3%) were male and the majority (79.1%) were aged 0–4 years. The mean age of children was 3.0 years (SD = 2.7). When

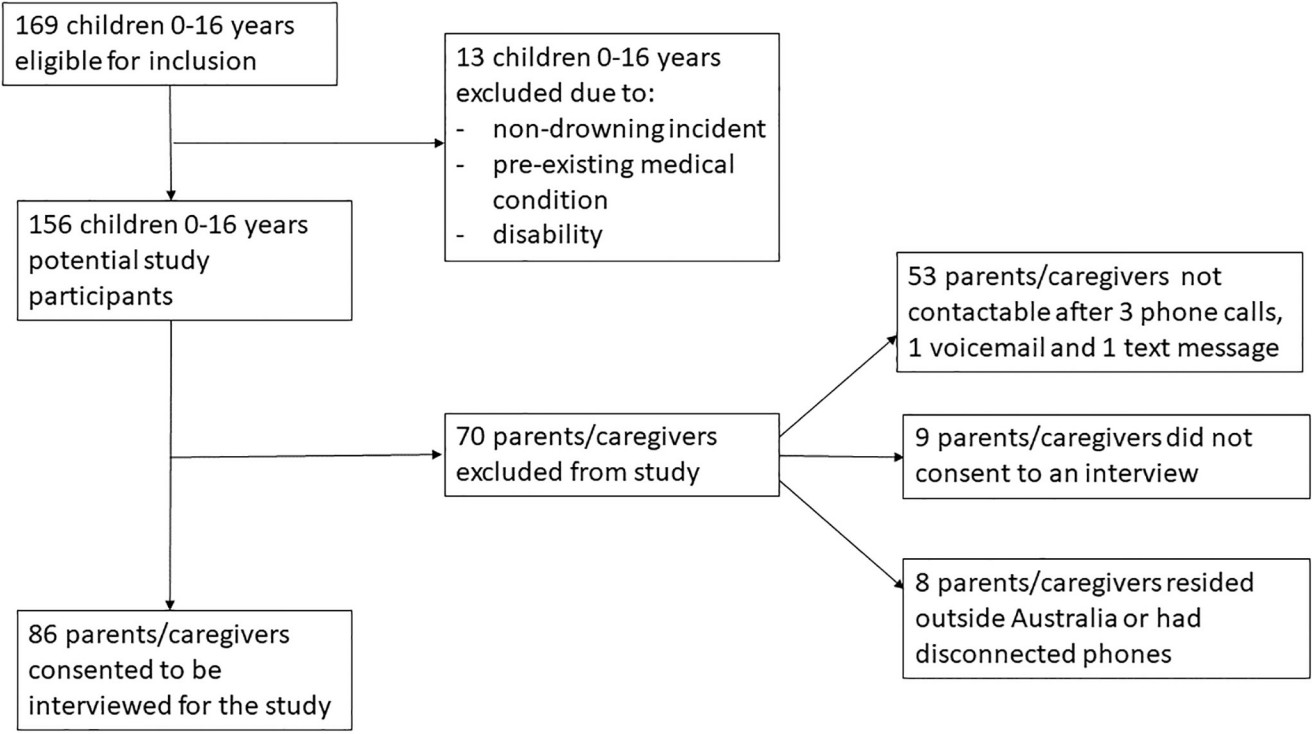

**Fig 1. Flow chart of participant eligibility and recruitment.**

**Table 2. Characteristics of interviewees and patients who experienced a drowning-related hospitalisation whose parents/caregivers were interviewed for the study.**

| | Number | % |
|---|---|---|
| Total | 86 | 100.0 |
| **Sex of interviewee** | | |
| Female | 74 | 86.0 |
| Male | 12 | 14.0 |
| **Relationship of interviewee to patient** | | |
| Mother | 72 | 83.7 |
| Father | 12 | 14.0 |
| Grandparent | 1 | 1.2 |
| Friend | 1 | 1.2 |
| **Sex of patient** | | |
| Female | 35 | 40.7 |
| Male | 51 | 59.3 |
| **Age group of patient** | | |
| 0–4 years | 68 | 79.1 |
| 5–10 years | 16 | 18.6 |
| 11–16 years | 2 | 2.3 |
| **Does patient identify as Aboriginal and/or Torres Strait Islander?** | | |
| Yes | 2 | 2.3 |
| No | 81 | 94.2 |
| No response | 3 | 3.5 |
| **Is a language other than English spoken at patient's home*** | | |
| Yes | 26 | 30.2 |
| No | 60 | 66.3 |
| No response | 3 | 3.5 |

*used to define if family are culturally and linguistically diverse (CALD).

compared to the New South Wales population as a whole the sample had a slightly lower proportion of Aboriginal and Torres Strait Islander people (2.3% in our sample vs. 2.9% in NSW) and a higher proportion of people who spoke a language other than English at home (30.2% vs. 22.0%) (Table 2).

Information about the drowning incidents can be found in Table 3. Almost half of all incidents occurred in summer (n = 38; 44.2%) and the most common time of day was the afternoon (n = 26; 30.2%). A swimming pool (n = 31; 36.0%), followed by a bathtub (n = 21; 24.4%) were the most common drowning locations. The presence of supervision was most commonly 'unsupervised and unaware' (n = 27; 31.4%) or 'unsupervised and aware' (n = 26; 30.2%). One to five minutes was the period of time that children were most commonly estimated to have been submerged (n = 41; 47.7%). A family member most commonly found the child (n = 73; 84.9%). Cardiopulmonary resuscitation (CPR) was commonly performed by parents (n = 29; 33.7%) and was not performed in 39 cases (45.3%). In almost half of all cases (n = 42; 48.8%) the child had not participated in water familiarisation/swimming lessons before the drowning incident (Table 3).

## Qualitative incident description

In coding the interviewees description of the incident, five themes emerged: lapse of supervision; unintended access; brief immersion; fall into water; and ongoing impacts.

**Table 3. Drowning incident details.**

|  | Number | % |
|---|---|---|
| Total | 86 | 100.0 |
| **Season of drowning incident** | | |
| Summer (December-February) | 38 | 44.2 |
| Autumn (March-May) | 11 | 12.8 |
| Winter (June-August) | 7 | 8.1 |
| Spring (September-November) | 30 | 34.9 |
| **Time of day of drowning incident** | | |
| Morning (6–10am) | 6 | 7.0 |
| Midday (10am–2pm) | 7 | 8.1 |
| Afternoon (2–6pm) | 26 | 30.2 |
| Evening (6pm–10pm) | 10 | 11.6 |
| No response | 37 | 43.0 |
| **Site of drowning incident** | | |
| Bathtub | 21 | 24.4 |
| Beach | 5 | 5.8 |
| Inland/protected water | 3 | 3.5 |
| Pond* | 5 | 5.8 |
| Public Pool | 19 | 22.1 |
| Swimming Pool** | 31 | 36.0 |
| Other | 2 | 2.3 |
| **Presence of supervision** | | |
| From a distance | 19 | 22.1 |
| Unsupervised and aware | 26 | 30.2 |
| Unsupervised and unaware | 27 | 31.4 |
| Within arms' reach | 14 | 16.3 |
| **Estimated length of submersion time** | | |
| 0–20 seconds | 26 | 30.2 |
| 21–59 seconds | 8 | 9.3 |
| 1–5 minutes | 41 | 47.7 |
| 6–10 minutes | 4 | 4.7 |
| Unknown | 7 | 8.1 |
| **Person who found child** | | |
| Family member (parent, caregiver, sibling, grandparent, or other relative) | 73 | 84.9 |
| Other (bystander, friend, lifeguard, swimming instructor) | 13 | 15.1 |
| **Administration of CPR** | | |
| Not performed | 39 | 45.3 |
| Parent/caregiver | 29 | 33.7 |
| Lifeguard | 5 | 5.8 |
| Neighbour/bystander | 5 | 5.8 |
| Other family member | 7 | 8.1 |
| Unknown | 1 | 1.2 |
| **Child participation in water familiarisation/swimming lessons before the incident?** | | |
| Yes | 42 | 48.8 |
| No | 37 | 43.0 |
| Not answered | 7 | 8.1 |

*Including garden ponds and fishponds.

** Note includes permanent, inflatable and portable pools.

**Lapse of supervision.** Lapses in supervision occurred in drowning incidents at a variety of aquatic locations. In several incident descriptions, the interviewee described small and seemingly benign distractions which impacted supervision, leading to a drowning incident. In several cases this involved care for other children: "*I was outside, my son and his sibling were inside when his sibling began to cry. I went inside and my son ran outside where he was unsupervised for 5 minutes. I had accidently left the pool gate open and I found my son head down in the pool. He received CPR from me and began to cry after two cycles of CPR*" (Father, child aged 3 years at time of incident, home swimming pool). In another case "*My son was at a public pool with me and my other child. I went a short distance to pack away lunch and attend to my other child briefly as my son continued to play in the water. I returned 3–5 mins later to find my son at the bottom of the pool. Unresponsive. There was an emergency doctor who saw the incident and performed CPR. Lifeguards also arrived and a helicopter transported my son to hospital*" (Mother, child aged 3 at time of incident, public swimming pool).

Household tasks such as retrieving towels while bathing children was also discussed as leading to a lapse in supervision leading to a drowning incident: "*I put my son in the bath with his older brother (4 years) and left the room to get towels/clothes. I was away for 2–3 minutes when my 4 year old son alerted me to my son being underwater and unable to get himself out. I pulled him out and he was red in the face and blue around lips. He coughed and was conscious. No CPR required. I took him straight to the doctors and then took an ambulance to hospital*" (Mother, child aged 1 year at the time of the incident, bathtub).

An absence of supervision can also occur in a formal setting, as one parent described their child's drowning incident after swimming lessons: "*My daughter had finished swimming lessons and was released early. Her dad was not there yet to get her however she wandered off and was found in the pool by another child. A child told their mother and lifeguards performed CPR. It is thought that our daughter was submerged for 1 minute*" (Mother, child aged 4 years at time of incident, public swimming pool).

Several incident descriptions provide powerful insights regarding the dangers of leaving children to supervise other children around the water: "*My child was playing with her sister in the backyard with a large bucket of water. The bucket was a tall laundry basket. They were both reaching into the bucket and playing. I was within arm's reach of my children. I looked away for one second to look at my phone. My husband then noticed that our child had fallen head first into the bucket and couldn't get herself out so he pulled her out. She would have been in water for about 5 secs. She was gasping for air. I quickly put her head down on an angle and hit her back hard a few times, at the back of her lungs. She started coughing and was unsettled. We called the ambulance. The ambulance took her to hospital but she was breathing and awake the whole time. Our older sibling later confessed that she pushed her sibling into the bucket*" (Mother, child aged 1 year at time of incident, other- bucket).

Another incident highlights how little young children understand about drowning risk: "*I was bathing my two children in the shower. My daughter was 4 months old and her brother was 2 years old. The 4 month old was in a bath seat on the floor of the shower and the 2 year old was standing. I went to get towels across the hallway (30 secs), then I heard the older child laughing. I came back to find the older child had removed the bath seat from underneath his sister and she was immersed in the water. My daughter was not moving and was blue. She was on her stomach, I pulled her out and hit her back, I called out to my husband, I gave her 2 breaths, my daughter started coughing up water and was then responsive. We drove her to hospital to make sure she was ok*" (Mother, child aged 4 months at time of incident, shower).

Even older children are not able to provide adequate supervision as this incident description details:

"*My son was playing in a friend's pool. I went to feed my newborn and left the pool area asking one of the teenagers to keep an eye on my son while I was away. I returned 3–5 minutes later to find my son floating face down in the pool and unresponsive. I performed CPR even though I had not been trained, until the ambulance arrived*"

(Mother, child aged 2 years at time of incident, home swimming pool).

**Unintended access.** Many incidents, particularly those in backyard swimming pools were coded to a theme of unintended access. Some incident descriptions highlighted potential weaknesses in pool barriers allowing for children to gain access to the water, such as "*Husband found son face down in the family pool after losing sight of him for up to 5 minutes. It appears our son followed the cat through a small gap between the boundary wall and the fence of the pool. My husband performed CPR and an ambulance was called*" (Mother, child was 1 year old at time of incident, home swimming pool).

In another case, multiple barriers failed to keep the child safe including a playpen, a door and the backyard pool fence: "*I put our son in a playpen in the living room to play while I was arranging the laundry in my bedroom. He was unattended for approximately 2.5 to 3 mins when I realised that he had tipped over the playpen and walked out through the open sliding door into the backyard. I noticed that the pool gate had been left open by my grandfather (with a hook) and that my son was floating in the pool on his back. I pulled him out of the pool and he was unconscious so I screamed for help. I attempted CPR. A neighbour came and administered CPR while I called an ambulance. He made some noises but was in and out of consciousness. The ambulance arrived and took him to our local hospital, he was then transferred to another hospital*" (Mother, child aged 1 year at time of incident, home swimming pool).

In other examples the barrier was non-existent: "*Our family was at a friends' place. I was putting clothes away upstairs, my son climbed down the stairs through the doggy door to the backyard and fell into the plunge pool which has no fence. I called out and started looking for him after 1–2 mins. I found him in the pool on his back, face submerged. I pulled him out, he was conscious, coughing up water, shocked, tired and exhausted. We have asked the owner to build a fence but he never did it and the family have since moved*" (Mother, child aged 1 year old at time of incident, home swimming pool).

In several other incident descriptions, children have breached the barrier by climbing. One interviewee said "*Our child was playing in the backyard unsupervised. He pulled the bike over to the edge of the home swimming pool fence, climbed the closed fence and jumped over and got access to the water. He was unsupervised for just 5 mins, then his grandma found him floating in the pool. I pulled him out and started CPR, after 10 compressions, he started breathing*" (Mother, child aged 2 years at time of incident, home swimming pool).

In another case, the incident description details the child sustaining an additional injury when climbing the pool fence which compounded the drowning risk: "*Our child climbed the pool fence and fell, knocking himself out and subsequently falling into the pool. I had left to make tea but returned shortly due to gut instinct. I gave him CPR and his pulse returned, however he was airlifted to hospital and intubated*" (Mother, child aged 5 year at time of incident, home swimming pool).

In one scenario, parents inaction to repair known issues with the pool fence, combined with a lapse of supervision, contributed to the drowning incident: "*Our son was asleep in the house and I was changing the nappy of my younger child, and my husband was inside the house also. Our son had woken up early on the Sunday morning and disappeared for a while. He is a very good climber. My husband found him in the swimming pool around 10:10 am, his face was*

*up at top of the pool, blue lips, not responding. He pulled him out and called an ambulance. My husband did chest compressions for about 1 minute and then he was responsive and opened his eyes, but his eyes were rolling. When ambulance came, they suctioned out the water and gave him oxygen. We knew that the pool fence lock had a problem and the latch wouldn't close, but we hadn't informed the landlord to come and fix it"* (Mother, child aged 2 years at time of incident, home swimming pool).

**Brief immersion.** Incidents coded to the brief immersion theme commonly involved infants slipping or falling into water while bathing or showering while parents/caregivers were in close proximity. As one interviewee said *"Our child was in the shower with me (his mum) and toppled forward into the water. His mouth and nose were briefly immersed in water. I picked up child who was red in the face"* (Mother, child was 11 months old at time of incident, shower).

Because of their young age, children were often taken to hospital out of an abundance of caution. As one interviewee said: *"My husband and I were bathing our daughter in the main bath. She was sitting un-aided (had just started sitting) and she fell face forward in the bath for 1–2 secs. My husband pulled her out and she was coughing, spluttering and crying. We googled what to do and called the health helpline as we were concerned about secondary drowning and they advised us to take her to hospital"* (Mother, child aged 7 months at time of incident, bathtub).

The risk of brief immersion is also present in small depths of water, such as in a baby bath. As one parent said, describing an incident in a baby bath: *"His father was washing the baby in small tub while the baby was on his stomach. He slipped into the tub with his face immersed for a few seconds. His father took him out of the water and patted his back and water came out from his mouth and nose. The baby was unresponsive for 30–40 seconds and then he was ok."* (Mother, child was aged 3 weeks at time of incident, baby bath).

**Fall into water.** Slipping or falling into water was another common theme in the narratives of drowning incidents in this study. These slips and falls included falls while unsupervised into the backyard pool: *"Child slipped off pool edge and into the water. Unwitnessed by adults. I was distracted for 1 minute and another 6 year old alerted me and I found her floating face up in the pool. I would estimate her time in the water to be between 30 seconds and 3 minutes. I took her out of the water, her lips were blue, she made gurgling noises but she became responsive within 10 seconds. The ambulance came later"* (Mother, child aged 5 years at time of incident, home swimming pool).

Such slips and trips are not restricted to pools, another interviewee discussed their child's drowning incident at a river location: *"Our child along with his friends (same age) were playing and running along the sand by the river while we were supervising them from a distance. Our child was running then fell into the water, I saw him fall. He started screaming and shouting. My husband ran and got him out of the water. When we pulled him out, he was not fully conscious. My husband and other friends performed CPR and then called an ambulance"* (Mother, child aged 7 years at time of incident, river).

**Ongoing impacts.** The theme of ongoing impacts commonly involved two sub themes: ongoing trauma and persisting health consequences as a result of the non-fatal drowning. With respect to ongoing trauma on family members, one interviewee described the drowning incident and aftermath as follows:"*[He] was in the pool with dad, uncle and sister and cousins for 30 minutes playing on flotation device. He fell off and went under water. During that time, dad had gone to the toilet and uncle did not see him. Dad returned to find him under water and pulled him out blue. Both dad and uncle performed CPR and he coughed and vomited immediately. Sister, was traumatised and blamed herself. Dad and uncle also experienced trauma from the incident."* (Mother, child aged 4 years at time of incident, public pool).

One interviewee described the emotions felt while their child underwent treatment post the drowning incident: *"One short moment can change your life, you can never be too safe. There is nothing worse than the feeling of helplessness and pain. You find yourself sitting and watching your child in a hospital bed knowing there is nothing else you can do to help this precious soul. All you have is prayer"* (Mother, child aged 1 year at time of incident, home swimming pool).

Interviews also captured ongoing trauma impacting the child involved. As one interviewee said *"Child was at the beach, swimming by herself while she was being supervised from a few meters away. The waves were dunking her in the water so she was trying to move away from the waves. One big wave hit her and she didn't come up. She was submerged for 20 secs to 3 mins. The lifeguards were alerted and got her out of the water. She was unresponsive, not breathing and blue. Lifeguards performed CPR and she started breathing but not responding. Ambulance called and she was taken to hospital. She was in the intensive care unit for 2 days. Now is back to normal but she mentions the accident 10 times a day—she is affected by it emotionally"* (Mother, child 8 years old at time of incident, beach).

Another sub-theme within the ongoing impacts theme, was that of persisting health consequences. As one interviewee said *"We were on a family holiday overseas, staying at a hotel. The whole family was at the hotel pool area, there were 5 children and 10 adults in or around the water. Child was swimming in the kid's pool with his snorkel on. He had used the snorkel before with no problem and he was a very confident swimmer. As parents we were out of the pool talking to others and grandparents were close by in the pool. After 5 mins or less, I realised that he wasn't around and his uncle noticed that he had gone down to the bottom of the pool with his snorkel on. We realised that he had swallowed water through the snorkel and drowned. His father got him out of the water and did CPR as well as a nurse. He woke up, was taken a local hospital and then back to Australia for ongoing care and rehab"* (Mother, child aged 5 years at time of drowning, hotel pool).

Another interviewee recounted the incident which left their child with serious health problems: *"We were at our home. Child went missing for 1 minute. Found unconscious and floating in the pool by parents. Father did CPR for 2 minutes. Police arrived prior to ambulance, and revived child. Child was put in lateral position due to vomiting. Taken to hospital in critical condition and admitted to the intensive care unit. We think that the child squeezed through the fence of the swimming pool. Child now has brain damage from the incident and is having ongoing rehab"* (Mother, child aged 1 year at time of drowning, home swimming pool). Although it was not specifically investigated in this study, four of the interviewees mentioned that as a result of the drowning incident, the child had sustained some form of brain injury (mild to severe).

## Qualitative parents/caregiver advice to others

There were five key themes identified within the parents/caregivers advice to others responses. These were importance of supervision, importance of pool barriers and maintenance, importance of CPR training and emergency response, that drowning is quick and silent and the importance of learning swimming and water survival skills.

**Importance of supervision.** When discussing supervision many interviewees highlighted the need for vigilance and also how easy it is for parents to become distracted usually by an everyday task such as getting something out of a bag, preparing food, attending to another child (in or out of the water), speaking to other adults/guests, other household tasks and checking mobile phones. One interviewee stated: *"Carers must be vigilant! Don't get chatting with people or distracted when a child is in water. You have to watch them all the time and be within arm's reach of them"* (Mother, 8 year old child at time of incident, beach).

Of distractions another interviewee said *"If a parent needs to get a drink or go to their bag while their child is in or around water, they should take the child with them because they could drown in a few seconds, even if you put floaties on them"* (Mother, child aged 2 years, public pool). Another interviewee highlighted difficulty in providing focused supervision when tired, saying: *"I was a bit tired and distracted so I want to advise parents to always have 100% concentration on your kids when they are around water. We were lucky, but we could have both drowned if no one helped us get out"* (Mother, child aged 2 years at time of incident, harbour).

This need for focused supervision was also highlighted in different aquatic environments such as a public or hotel pools *"Parents or carers need to be extra vigilant when in and around public pools or hotel pools because you can lose sight of your child. Don't assume that someone is looking after your child when you are in a big group at the pool. Make sure you know who is watching your child. . ."*(Mother, child aged 5 years hotel pool) and bathtubs *"Parents should be organised before they bathe their kids, so you don't have to leave the room when they are in water. Have towels ready"* (Mother, child aged 4 months at the time of the incident, baby bath).

Similarly, another interviewee highlighted the need to watch children when they are around water, even if they are not intended to get in the water *"Always watch your kids when they are AROUND water because they can fall in any time. Sometimes they don't listen to parents and have so much energy and are quick"* (Mother, child was 7 years old at time of incident, river).

Another interviewee spoke of the importance of proximity when supervising and not relying on others to provide supervision *"Get in the water with your child/ren! Unless there is an adult there supervising, they are not safe! You can't take your eyes off children when they are in water because drowning happens so quick. I thought that the other children in the pool would look out for him, but they didn't"* (Mother, child aged 3 at time of incident, home swimming pool).

**Importance of pool barriers and maintenance.** One of the key themes in the parent recommendations to other parents was the importance of pool barriers and maintenance. As one interviewee said: *"Always check that the pool gate is firmly closed before you leave the pool area, even if it is for a few seconds"* (Father, child 3 years old home swimming pool).

In hindsight, parents/carers acknowledge the importance of an effective pool barrier even though some were aware of faults or misuse of a barrier before a drowning incident *"If your gate is faulty or broken, fix it as soon as possible. And if it's a rental property, notify your real estate or landlord straight away"* (Mother, child aged 3 at time of drowning, home swimming pool).

Another interviewee identified the importance of removing climbable objects from near the pool fence: *"If your child is a climber, move away any chairs, bikes, toys and anything that the child can climb up on and climb over the pool fence or to the source of water"* (Mother, child aged 2 at time of incident, home swimming pool).

Another interviewee highlighted the challenges tenants can face when renting a property with a pool, *"If someone has a pool, they need to make sure the gate has a working lock and to ask the owners of the rental property to fix it if it has a problem. Also, to make sure there is nothing around the fence that a child can climb over and get to the pool themselves. I urge all landlords take responsibility to make sure the lock of their pool gates are working before they give it for lease"* (Mother, child aged 2 years old home swimming pool).

**Importance of CPR training and emergency response.** One of the key themes in parent recommendations to other parents was how important it was to know how to perform cardiopulmonary resuscitation (CPR). As one interviewee said *"knowing CPR is very important. All parents should do the training; it saved my child's life!"* (Father, child aged 1 month at time of drowning, bathtub). One interviewee noted *"It should be compulsory for parents to do a free CPR training when they have their first child"* (Mother, child aged 1 year old, backyard pond).

A number of the narratives describing a drowning incident noted that although they had learnt CPR skills, they were unable to perform those skills when faced with the situation of their own child needing resuscitation. One interviewee said *"Every parent should know CPR and needs to be confident in it. I am so shocked and couldn't remember what to do, luckily my stepdad jumped in and attempted some sort of CPR"* (Mother, child aged 1 year old, backyard pond).

A few interviewees noted that *"CPR training should be more available, approachable and affordable for parents to do"* (Mother, child was aged 3 at time of incident, home swimming pool).

**Drowning is quick and silent.** Many respondents commented on the speed with which drowning occurred. One interviewee said *"I didn't know drowning can happen so quick! Never leave a child near water on their own, even for a minute! My son is very familiar with water and is a good swimmer, yet he still almost drowned!"* (Mother, child aged 2 years at time of incident, home swimming pool). Another interviewee commented that drowning didn't look like they expected it to *"Everyone should learn and understand the signs of drowning. I always thought that drowning would be frantic and that the child would scream and shout, but it was in fact very quiet and quick. There were adults in the water but they didn't help my child–maybe because they didn't know the signs of drowning"* (Mother, child aged 4 years at time of incident, hotel pool).

Interviewees also commented about the lack of sound when a child is drowning: *"Drowning happens so quickly and it's so quiet, there was no splashing, no bubbles, no sound"* (Mother, child 2 years old at time of incident, swimming pool). Another said: *"People think that when a child is drowning they would scream out, but it's not like that. They often make the action of climbing a ladder, and their head bops up and down. Adults and children should be educated on the signs to look out for when a child is drowning"* (Mother, child aged 6 years at time of incident, home swimming pool).

**Importance of learning swimming and water survival skills.** The importance of learning swimming and water survival skills was one of the key themes in parent recommendations to other parents. As one interviewee said: *"Children should learn how to confidently swim and not panic if they fall in water–private lessons would be better than school lessons"* (Mother, 9 years at time of incident, lake).

An interviewee identified the importance of swimming lessons for adults and children alike, saying *"It's important for migrants like myself who moved here after high school years and who have not had swimming lessons to learn water safety, to learn to swim and to know the dangers of water–not to take it lightly"* (Mother, child aged 2 years at time of incident, harbour).

Although participating in swimming lessons were identified as an important advice for parents and carers of young children, one interviewee highlighted some of the challenges facing children including fear of water after a non-fatal drowning incident: *"I encourage all parents to take your children to swimming lessons. This is very important in Australia. I have been trying to encourage my son to do lessons after the incident, but he is now scared"* (Father, child aged 7 at the time of the incident, public pool).

## Discussion

Drowning is a leading cause of injury-related mortality and morbidity in Australia, particularly for young children and adolescents [2]. Epidemiological data has enhanced understanding of risk factors for drowning, however there is a dearth of qualitative data on non-fatal drowning. The present study was unique in its methodology of collecting data on the circumstances surrounding the non-fatal drowning incident via semi-structured in-depth interviews with

parents and carers of children who experienced a drowning incident with or without morbidity. Retrospectively interviewing the families of the participants to gather more comprehensive qualitative data than that which is available in the medical database and clinical notes adds depth to the understanding of the circumstances surrounding the drowning incident. Contacting the families three months after the drowning incident allowed the parent/carer to have some time to process the incident and provided enough time for them to be emotionally ready for the interview. This study also catered to languages other English and given the diverse populations which the included hospitals service, results include 34% of children from households which a language other than English was spoken. Given the unique drowning risks facing migrant and culturally and linguistically diverse families face, and the limited literature on this topic [12], this is a strength of the current study.

We present a discussion of the key findings of this research themed around the four key strategies for reducing child drowning: supervision, restricting access to water such as by using pool barriers, swimming lessons and cardiopulmonary resuscitation and emergency response.

## Supervision

Active adult supervision is one of the most effective drowning prevention strategies and is regularly communicated to parents and carers of young children via media and advocacy campaigns [13]. However, this study identified the majority of children who had a non-fatal drowning incident were not supervised at the time. Interviewees spoke of the challenges in maintaining vigilance and focus on young children in and around water when facing competing demands and distraction from everyday tasks. Similar causes of distraction have been noted in cases of fatal drowning among young children in Australia, highlighting the need for communication about accepted forms of distraction (such as mobile phones and socialising) as well as lesser known forms of distraction such as indoor and outdoor household chores, or confusion regarding supervision duties within a group [14, 15]. In addition, many caregivers mentioned the issue of a brief immersion, which is similar to studies of fatal child drowning in Australia where time left unsupervised is recorded in coronial data [14, 16]. Despite the perceived short period of time the child was immersed in water, such incidents resulted in children being admitted to hospital, no doubt causing significant stress on child and caregivers. Such findings are an important reminder of the need to prevent the immersion incident in the first place, including through effective parental or caregiver supervision [13]. Drowning prevention organisations and advocates should consider the inclusion of parental and caregiver information about the dangers of even a brief immersion, and the importance of supervision as a primary preventive strategy.

## Restricting access to water via pool barriers

The most common place children had a non-fatal drowning incident was in swimming pools (36%), which is mirrored in fatal drowning data as the location of greatest risk for young children in Australia [3, 14, 17]. Interviewees identified leaving pool gates propped open in a third of cases, as well as also mentioning the risk of having climbable objects in the pool area or near the barrier. Although such advice is regularly the focus of child drowning prevention media and advocacy campaigns [13], these risk reduction strategies clearly warrant ongoing communication to parents and caregivers of young children.

Additionally, several interviewees discussed the pool safety challenges facing tenants renting properties with a pool, imploring landlords to ensure the pool barrier is safe prior to renting a property or responding to maintenance requests as needed. Such findings reinforce the benefits of regular inspection regimes at point of sale and lease, such as is in place in New

South Wales [18], however, also indicates a need for further education of tenants regarding their rights around pool barrier compliance and for landlords and leasing agents with respect to a landlord's responsibility when it comes to pool safety.

## Learning swimming and water survival skills

In-depth detail regarding the child's swimming ability and previous participation in swimming lessons gathered through this study, differs from coronial data where such information is rarely recorded [3]. In the present study, parents and caregivers reported half of the children who had a non-fatal drowning incident had previously had swimming lessons or were described as being familiar with water. This finding is an important reminder to parents and caregivers to not assume children have the required level of skill to keep themselves safe in the water and reiterates the concept of swimming lesson as one in a range of strategies to reduce drowning risk [13]. A further 43% of children were described as not having previously had swimming or water familiarisation lessons. Given that children from culturally and linguistically diverse backgrounds accounted for 34% of incidents investigated in this study, and migrants to Australia have been identified as being less likely to have accessed water familiarisation classes for their children under five years of age [19], specific and culturally appropriate promotion of swimming lessons to migrant families is warranted. It is important also for parents and adult caregivers to also learn how to swim and know what to do in a drowning emergency. Swimming lessons for both adults and children should be conducted in a culturally appropriate way, with swimming teachers from the same culture and language wherever possible.

Although not the intent of this research, several interviewees discussed the ongoing trauma and psychological effects of non-fatal drowning on themselves, their families, and the child. In a number of interviews, the child's persistent fear as a result of the non-fatal drowning has been a barrier to enrolment or re-enrolment in swimming lessons. Research has also shown that a bad experience around water can impact a child's ability to learn to swim once enrolled [20].

## Cardiopulmonary resuscitation (CPR) training and emergency response

The vast majority of children who drowned were found by a family member, as is commonly reported in coronial data [16], supporting the notion that parents and caregivers should be trained in CPR. An important finding of the interviews was the stress of the drowning incident rendering some interviewees unable to provide CPR to their own child. In fact, 45% of children who drowned in the incidents included in this study did not receive CPR at all. Additionally, several of the incident narratives mentioned administering blows to the child's back after they had been retrieved from the water. Given the provision of CPR and the speed with which this occurs are important predictors of outcome after drowning [21], these are concerning findings. To address the specific barriers to participation, and needs, of culturally and linguistically diverse families, it is recommended that CPR courses be run in language, with specific in-language promotion of courses to the culturally and linguistically diverse populations.

## Strengths and limitations

This study adds to the extremely limited qualitative research exploring non-fatal drowning. The study comprises the experiences of parents and caregivers of both young children and adolescents from metropolitan, regional and rural areas of New South Wales, Australia. Similarly, if needed, interviews were conducted in languages other than English thus avoiding a bias towards English speaking participants only. The findings of this study must be considered however, in light of some limitations. Recall bias or social desirability bias may have impacted

parents and caregivers resulting in a lack of detail being provided regarding pool barriers and overall compliance [22, 23]. Time left unsupervised was estimated by parents and is likely to be an underestimation [16]. There were just two incidents in portable pools, believed to be an under representation of the proportion of portable pools in use in the community. Injury severity score (ISS) was not used in the analysis as it is not always provided in the medical records. This means there was no assessment of the severity of the non-fatal drowning-related hospitalisations that parents and caregivers were interviewed about. In the results there were a number of cases which are best described as brief immersions, where the child slipped from a parent's grip and was momentarily underwater. Many of the hospital attendances for these incidents appear to have been precautionary, which has been previously found in non-fatal drowning research, particularly for children under five years of age [24]. Thirty-four percent of children who drowned were from culturally and linguistically diverse backgrounds and while families who did not speak English fluently were catered for through the use of language interpreters, the interview questionnaire was not detailed enough to give good insights into the factors associated with drowning incidents for culturally and linguistically diverse families. Although language spoken at home was collected, it remains unclear if speaking a language other than English was a factor in the drowning incident. Additionally, children with a pre-existing health condition or disability were excluded. Given children with conditions such as epilepsy and autism spectrum disorder can be at increased risk of drowning [25, 26], further research is recommended with parents/caregivers of such children in the future. Similarly, just 2% of those interviewed were describing non-fatal drowning incidents of adolescents aged 11–16 years. Although children under five are the age group most at risk of drowning, recent research indicates adolescent drowning rates are increasing in Australia [17]. Therefore, it is recommended that further research on experiences of non-fatal drowning among adolescents be conducted in the future.

## Conclusion

Using a qualitative approach, this study has identified powerful insights regarding causal factors implicated in cases of child non-fatal drowning from those directly impacted. Study findings reconfirm that children under five years are at the greatest risk of drowning, that drowning most commonly occurs in a home swimming pool or bath, and that children who are unsupervised are more likely to drown than children who are closely and actively supervised while in or near water. Supervision, effective pool barriers, learning swimming skills and learning CPR remain key child drowning prevention strategies, however consideration needs to be given as to how to best communicate this to parents and the wider community, particularly culturally and linguistically diverse families, as the absence of these factors remain the primary causes of both fatal and non-fatal drowning in children.

## Supporting information

**S1 File. Questionnaire to be completed by researcher (based on patient records).**
(DOCX)

**S2 File. Interview schedule with parent/carer.**
(DOCX)

## Acknowledgments

The authors would like to acknowledge the following people for assistance with the project: Kids Health Promotion Unit, Sydney Children's Hospitals Network: Candace Douglass,

Dushyanthi Nagaratnam, Erin Collimore, Louella Monaghan, Stacie Powell and Pamela Lopez-Vargas; Centre for Trauma Care, Prevention, Education and Research: Professor Daniel Cass, Patricia Manglick, Frank Ross, Carla Ghisla and Laura Holliday; Management Support and Analysis Unit (SCHN): Babita Banerjee; Trauma Service at Sydney Children's Hospital: Sarah Adams, Nevin William and Claire Collins; John Hunter Children's Hospital: Teagan L. Way and Julie Evans; NSW Ministry of Health Centre for Epidemiology and Evidence: Melissa Irwin; Royal Life Saving Society NSW; and Ministry for Police and Emergency Services (2012–2014).

## Author Contributions

**Conceptualization:** Boshra Awan.

**Data curation:** Boshra Awan, Suzanne Wicks.

**Formal analysis:** Boshra Awan, Suzanne Wicks, Amy E. Peden.

**Investigation:** Boshra Awan.

**Methodology:** Boshra Awan, Suzanne Wicks.

**Project administration:** Boshra Awan.

**Visualization:** Amy E. Peden.

**Writing – original draft:** Boshra Awan, Suzanne Wicks, Amy E. Peden.

**Writing – review & editing:** Boshra Awan, Suzanne Wicks, Amy E. Peden.

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
