## [Decision Letter · Decision Letter 0]

9 Sep 2022

PONE-D-22-17354A qualitative examination of causal factors and parent/caregiver experiences of non-fatal drowning-related hospitalisations of children aged 0-16 yearsPLOS ONE

Dear Dr. Peden,

Thank you for submitting your manuscript to PLOS ONE. After careful consideration, we feel that it has merit but does not fully meet PLOS ONE’s publication criteria as it currently stands. Therefore, we invite you to submit a revised version of the manuscript that addresses the points raised during the review process.

We look forward to receiving your revised manuscript.

Kind regards,

Senthil Kumaran, MBBS, MD, DNB

Academic Editor

PLOS ONE

Journal Requirements:

2. Please provide additional details regarding participant consent for your retrospective study. In the ethics statement in the Methods and online submission information, please ensure that you have specified what type you obtained (for instance, written or verbal, and if verbal, how it was documented and witnessed). If your study included minors, state whether you obtained consent from parents or guardians. If the need for consent was waived by the ethics committee, please include this information.

Additional Editor Comments:

Comment:

Abstract and Introduction

1. It is unclear what ICD is referring to in the Abstract.

2. The wording in the Abstract of “are to be made” is awkward and should be replaced with wording such as “should be made”?

3. In the Introduction, a stronger rationale for the role of a qualitative approach in this context should be offered in addition to simply its absence historically (p. 3).

Method and Results

4. Why were children with a disability or pre-existing health conditions in particular excluded from the study? (p. 5). Please provide a rationale.

5. “pre-exiting” should re “pre-existing” on p. 5.

6. Can more information please be provided about the consent process for parents being contacted? Did the parents agree to be contacted for research purposes as part of the care they received at the hospitals or was a waiver of consent received in this instance to cover the screening process of hospital records? Given the sensitive nature of the topic among a potentially vulnerable cohort, more detail about this process would be useful for the reader (p. 5).

7. What is meant by “to.. ultimately ensure successful recruitment” as it sounds a little coercive with the current wording? Perhaps re-word? (p, 5).

8. The qualitative analysis undertaken was fairly superficial but likely appropriate given the research question of interest and the fact the study did not comprise an indepth exploration of the effects of the likely traumatic experience on parents/caregivers.

Discussion

9. Could the Discussion include slightly more consideration of the result relating to brief immersions and the ramifications for training/education strategies?

10. Would it be useful to draw more comparisons of the findings of the present study with those that have examined fatal drownings re identified similarities/differences?

Reviewers' comments:

Reviewer's Responses to Questions

**Comments to the Author**

1. Is the manuscript technically sound, and do the data support the conclusions?

Reviewer #1: Yes

2. Has the statistical analysis been performed appropriately and rigorously? 

Reviewer #1: Yes

3. Have the authors made all data underlying the findings in their manuscript fully available?

Reviewer #1: No

4. Is the manuscript presented in an intelligible fashion and written in standard English?

Reviewer #1: Yes

5. Review Comments to the Author

Reviewer #1: The manuscript PONE-D-22-17354 describes a qualitative study exploring parent/caregiver experiences of non-fatal drowning related hospitalisations of their children. The major strength of the paper is that it examines the experiences of parents for an important supervision behaviour in efforts to inform future preventative strategies. The main issues I have with the paper lie in (1) an absence of detail of the ethical procedures followed for a sensitive topic among a vulnerable group and (2) the lack of rationale for exclusion of children with a pre-existing health condition/disability. These minor points, along with other issues, are detailed below.

Abstract and Introduction

1. It is unclear what ICD is referring to in the Abstract.

2. The wording in the Abstract of “are to be made” is awkward and should be replaced with wording such as “should be made”?

3. In the Introduction, a stronger rationale for the role of a qualitative approach in this context should be offered in addition to simply its absence historically (p. 3).

Method and Results

4. Why were children with a disability or pre-existing health conditions in particular excluded from the study? (p. 5). Please provide a rationale.

5. “pre-exiting” should re “pre-existing” on p. 5.

6. Can more information please be provided about the consent process for parents being contacted? Did the parents agree to be contacted for research purposes as part of the care they received at the hospitals or was a waiver of consent received in this instance to cover the screening process of hospital records? Given the sensitive nature of the topic among a potentially vulnerable cohort, more detail about this process would be useful for the reader (p. 5).

7. What is meant by “to.. ultimately ensure successful recruitment” as it sounds a little coercive with the current wording? Perhaps re-word? (p, 5).

8. The qualitative analysis undertaken was fairly superficial but likely appropriate given the research question of interest and the fact the study did not comprise an indepth exploration of the effects of the likely traumatic experience on parents/caregivers.

Discussion

9. Could the Discussion include slightly more consideration of the result relating to brief immersions and the ramifications for training/education strategies?

10. Would it be useful to draw more comparisons of the findings of the present study with those that have examined fatal drownings re identified similarities/differences?

Recommendation:

The stated points above are easily addressed and the manuscript is a worthwhile contribution to the extant literature. The paper contributes usefully to our understanding of this important parental supervision/child safety topic.

6. PLOS authors have the option to publish the peer review history of their article (what does this mean?). If published, this will include your full peer review and any attached files.

Reviewer #1: No

---

## [Author Response · Author response to Decision Letter 0]

25 Sep 2022

26-09-2022

Senthil Kumaran, MBBS, MD, DNB

Academic Editor

PLOS ONE

Dear Senthil Kumaran, 

We thank the reviewer for their work in providing feedback to our manuscript PONE-D-22-17354 entitled “A qualitative examination of causal factors and parent/caregiver experiences of non-fatal drowning-related hospitalisations of children aged 0-16 years” which we submitted for consideration for publication in PLOSONE. 

We are pleased to see there is support for our study after revisions. We have indicated below our responses to the reviewer’s comments in the uploaded file entitled "response to reviewers" and also made changes in the manuscript using the track changes functionality in Microsoft Word. 

We feel our manuscript is now stronger as a result of these revisions and we hope our manuscript is now ready for publication. It would be an honour to see our work published in PLOSONE. 

Yours sincerely, 

Boshra Awan

Kids Health Promotion Unit, 

Sydney Children’s Hospitals Network

boshra.awan@health.nsw.gov.au

---

## [Editor Report · Decision Letter 1]

6 Oct 2022

A qualitative examination of causal factors and parent/caregiver experiences of non-fatal drowning-related hospitalisations of children aged 0-16 years

PONE-D-22-17354R1

Dear Dr. Peden,

We’re pleased to inform you that your manuscript has been judged scientifically suitable for publication and will be formally accepted for publication once it meets all outstanding technical requirements.

Kind regards,

Senthil Kumaran, MBBS, MD, DNB

Academic Editor

PLOS ONE
---

## [Editor Report · Acceptance letter]

19 Oct 2022

PONE-D-22-17354R1 

A qualitative examination of causal factors and parent/caregiver experiences of non-fatal drowning-related hospitalisations of children aged 0-16 years 

Dear Dr. Peden:

I'm pleased to inform you that your manuscript has been deemed suitable for publication in PLOS ONE. Congratulations! Your manuscript is now with our production department. 

Kind regards, 

on behalf of

Dr. Senthil Kumaran 

Academic Editor

PLOS ONE